# Development of a Real-Time Recombinase Polymerase Amplification Assay for the Rapid Detection of African Swine Fever Virus Genotype I and II

**DOI:** 10.3390/pathogens11040439

**Published:** 2022-04-05

**Authors:** Titov Ilya, Sezim Monoldorova, Shin-Seok Kang, Seungri Yun, Hyeon-Seop Byeon, Nefedeva Mariia, Bo-Young Jeon

**Affiliations:** 1Department of Biomedical Laboratory Science, College of Software and Digital Healthcare Convergence, Yonsei University, Wonju 26493, Korea; titoffia@yandex.ru (T.I.); sezima91@gmail.com (S.M.); newstonek@naver.com (S.-S.K.); yunseungri98@gmail.com (S.Y.); 2Laboratory of Molecular Genetic Research, Federal Research Center for Virology and Microbiology, 601125 Volginskii, Russia; info@ficvim.ru; 3Chungbuk Veterinary Service and Research, Cheongju 28153, Korea; vetmed97@korea.kr; 4One Health Frontier Inc., Wonju 26493, Korea

**Keywords:** African swine fever virus (ASFV), recombinase polymerase amplification (RPA), CP204 gene, genotype I, genotype II

## Abstract

African swine fever (ASF) is a contagious viral disease in pigs and wild boars which poses a major threat to the pig industry. Rapid and accurate diagnosis is necessary to control ASF. Hence, we developed a rapid diagnostic method using a recombinase polymerase amplification (RPA) assay targeting the conserved sequences of *CP204L* (p30) thatcan rapidly detect ASF virus (ASFV) genotype strains I and II. The lower detection limit of the real-time RPA assay was 5 × 10^1^ copies per reaction. The real-time RPA assay effectively detected ASFV isolates and clinical specimens belonging to ASFV genotypes I and II. The sensitivity and specificity of the assay were 96.8% (95% confidence interval (CI): 83.3–99.9) and 100% (95% CI: 88.4–100.0), respectively. The agreement between the real-time RPA assay and a reference commercial real-time quantitative polymerase chain reaction (qPCR) was 100%. The real-time RPA assay had a detection time of 6.0 min (95% CI: 5.7–6.2), which was significantly shorter than that of qPCR (49 min; 95% CI: 47.4–50.6; *p* < 0.001). Thus, the developed real-time RPA assay is a rapid and accurate diagnostic tool for detecting ASFV genotypes I and II.

## 1. Introduction

African swine fever (ASF), a contagious disease in pigs and wild boars, is caused by the ASF virus (ASFV), which is a large double-stranded DNA virus belonging to the genus *Asfarviridae*. As the mortality rate among pigs affected by ASF is 100%, this disease poses a great threat to porcine husbandry. Moreover, no effective commercial vaccines are available for this disease. ASF continues to spread across European and Asian countries despite the implementation of preventive measures [1]. The disease has already spread across Germany and is transmitted among wild boars [2]. Furthermore, wild boar migration has led to the spread of ASF to other European countries. ASF outbreaks were reported in previously unaffected areas of Liguria and Piedmont, Italy in 2022, which may have been transmitted by wild boars, where the ASFV genotype II was detected. This further raises serious concerns about the spread of ASF in Europe. The situation in Asian countries is worse than that in European countries. Since the first outbreak of ASF in China in 2018, the disease has spread across China [3], Vietnam, Cambodia, Laos, Thailand, and the Republic of Korea, greatly affecting the agricultural industry [4]. In the Republic of Korea, since the first outbreak of ASF in pig farms near the demilitarized zone in September 2019, ASF outbreaks have continued to occur in Hwacheon in 2020 and Yeongwol in 2021 [5]. ASF appears to have been transmitted through wild boars in forested areas, and the occurrence of additional outbreaks is only a matter of time [6].

Thus, rapid and efficient detection of diseased pigs is important because the diagnosis-based implementation of appropriate and timely measures has a significant impact on quarantine management or surveillance in specific areas. In particular, the success of control measures depends on the availability of appropriate diagnostic tests that are both fast and accurate. Additionally, the interpretation of ASF diagnostic results could be complicated depending on the clinical presentation of ASF, the epidemiological situation, and the characteristics of the epidemic virus [7]. Another important requirement for the diagnosis of ASF is that the results should be available rapidly. Real-time quantitative polymerase chain reaction (qPCR) is a widely used molecular diagnostic method, which has the advantages of high sensitivity and accuracy. However, it requires expensive equipment and may take ~2–3 h to deliver results. Thus, it is suitable for specialized laboratories equipped with professional facilities. Therefore, there is a need to develop a rapid, reliable, and sensitive diagnostic method that can be used in fields.

The recombinase polymerase amplification (RPA) assay is a promising isothermal PCR method. It is a novel method using recombinase and isothermal polymerase, and its results can be obtained within 20 min without the need for thermal cycling [8]. Furthermore, because the RPA assay requires high-quality specimens, it can be used as an on-site test using a specific probe for amplified products [9]. Most PCR-based methods for detecting the ASFV genome are based on the amplification of the *B646L* gene, which encodes the outer capsid protein of p72 [10,11,12,13,14]. The *B646L* gene is used as a marker for ASFV genotyping because of its high number of mutations. Therefore, it is necessary to identify conserved target genes for the detection of ASFV. The *CP204L* gene encodes p30, a 30 kDa phosphoprotein of ASFV, which is involved in the internalization of the virus into the cell. p30 is released both at an early stage [15] and during the latency phase of ASFV infection [16]. It is one of the most immunogenic proteins for detecting ASFV infection and a suitable target for diagnostic assays [17]. We found that the *CP204* gene is highly conserved among the currently prevalent ASFV strains, including genotypes I and II, and that it could be utilized to detect ASFV.

Therefore, we developed a diagnostic method that can detect ASFV using the RPA method to diagnose ASF quickly and effectively by targeting the *CP204L* (p30) gene. We evaluated its performance using ASFV isolates and clinical specimens from pigs experimentally challenged with ASFV.

## 2. Results

### 2.1. Analytical Sensitivity of Real-Time RPA Assay

Appropriate primers and probes for ASFV detection were selected in a pilot study for real-time RPA. The primer pairs and probes used in the subsequent experiments are listed in Table 1. The synthesized double-stranded DNA fragment, which corresponded to the amplification target sequence of the *CP204L* gene, was used as a standard to determine the detection limit of the real-time RPA assay for ASFV. The synthesized DNA standard was serially diluted in 10-fold increments to achieve DNA concentrations ranging from 5 × 10^6^ to 5 × 10^0^ copies/μL. Based on the results of three independent real-time RPA assays with the diluted DNA standard, the detection limit of the real-time RPA assay was found to be 5 × 10^1^ copies per reaction, which could be detected within 12 min (684 s) (Figure 1).

### 2.2. Performance of the Real-Time RPA Assay on ASFV Specimens

Of the 31 ASFV specimens, 11 ASFV isolates were tested; all ASFV isolates were positive in the real-time RPA assay (Table 2). The mean detection time for the real-time RPA assay was ~6 min (364 s) (data not shown).

The performance of the real-time RPA assay was tested using specimens from pigs challenged with virulent strains of ASFV. Of the four pigs challenged with ASFV Volgograd-v (ASFV genotype II), three specimens collected 3-days post-infection were detected as positive in the real-time RPA assay, and the average detection time of the positive results was ~7 min (435 s). After that time point, specimens collected from pigs were positive in the real-time RPA assay, and the detection time was slightly decreased to ~5–6 min (316–360 s) (Figure 2). In addition, blood specimens obtained from pigs challenged with ASFV Congo-v (ASFV genotype I) were used for analysis. Specimens collected from pigs 3-days post-challenge were detected as positive in the real-time RPA assay, and the average detection time was 5.5 min (330 s). For specimens collected at 5- and 7-days post-challenge, all were detected as positive in the real-time RPA assay, and the detection time was ~5–6 min. Of the 31 ASFV specimens, including ASFV isolates and clinical samples from infected pigs, 30 (96.8%) were positive in the real-time RPA assay (Table 2).

ASF specimens were also analyzed using a commercially available reference qPCR kit (VDx ASFV qPCR kit) (Figure 3). Of the 31 ASFV specimens, 30 (96.8%) were positive in the qPCR analysis, which was identical to that of the real-time RPA assay (Table 2 and Table 3). Interestingly, the real-time RPA assay and qPCR yielded a negative result for the same specimen, presumably because the virus concentration was lower than the detection limit in the pig challenged with ASFV at an early stage of infection. Both assays yielded negative results for the other tested viruses, with a specificity of 100%. Therefore, the concordance rate between the two assays was 100% (Table 3).

The mean detection time of qPCR was 49 min (95% CI: 47.4–50.6), whereas that of the real-time RPA assay was 6 min (95% CI: 5.7–6.2). Therefore, compared to qPCR, the real-time RPA assay could detect ASFV significantly faster (*p* < 0.001). Furthermore, the real-time RPA assay could effectively detect both ASFV genotypes I and II not only in clinical isolates but also in specimens derived from infected animals. Together, these findings suggest that the real-time RPA assay can be used to detect ASFV in a variety of specimens.

## 3. Discussion

qPCR has been used as a standard method for ASF diagnosis owing to its excellent sensitivity and specificity for detecting ASFV in clinical specimens derived from domestic pigs and wild boars [7]. However, qPCR has limitations in its application to on-site diagnosis because it requires expensive equipment and prolonged time to yield results. In recent years, several isothermal DNA amplification methods have been developed as simple and rapid alternatives to PCR-based amplification for detecting various animal pathogens [18,19,20,21,22,23]. The RPA assay does not require initial heating for DNA denaturation and the results can be obtained within 20 min. There are few or no non-specific reactions that can be problematic in other isothermal amplification methods, such as loop-mediated isothermal amplification. In addition, the RPA assay can be applied to a wide range of specimens because it shows tolerance to PCR inhibitors, which may be present in some samples [24].

In the present study, we developed a real-time RPA assay based on an exo-probe for the rapid and sensitive detection of ASFV genotypes I and II. ASFV genotype II circulates in many countries and ASFV genotype I was recently reported in China [25]. Several PCR-based methods have been developed to identify the ASFV genome; however, most are based on the amplification of the B646L gene fragment, which encodes the outer capsid protein of ASFV (p72) [10,11,12,13,14]. The B646L gene was also used to determine the genotype of ASFV, indicating the presence of a genotype-specific mutation in the B646L gene. However, we selected the CP204L gene as an amplification target because it is highly conserved among ASFV strains belonging to genotypes I and II. Additionally, we found sequences that are highly conserved and contain few nucleotides variations in the CP204L gene of ASFV strains, including the representative strains of genotypes I and II, the “Congo”and“Georgia/2007” strains, respectively, as well as other genotypes, and designed primers and probes for these sequences. We found that a real-time RPA assay using the prepared primers and probe could effectively detect ASFV isolates and clinical specimens of ASFV genotypes I and II. The detection limit of the RPA targeting CP204L (p30) developed in this study was 50 copies per reaction. In contrast, in the case of other RPAs developed with B646L (p72) as a target, the detection limit was 93.4–150 copies per reaction [11,12,13]. In light of these results, the RPA developed in the present study was much more sensitive than the existing ones. The developed RPA showed 100% sensitivity for the ASFV isolates, 96.8% sensitivity for the ASFV-challenged specimen, and 100% concordance when compared to the World Organization for Animal Health (OIE) standard qPCR. The sensitivities of other RPAs ranged from 92.3% to 100%, which were calculated based on the OIE qPCR results for suspected blood, tissue, or environmental specimens but not including ASFV isolates as a standard reference [10,11,12,13]. However, our study could be considered a meaningful advancement because ASFV isolates were used as a reference and the results are shown for the specimens from ASFV-challenged pigs according to the selected time point post-challenge. Although this real-time RPA assay was designed to detect known ASFV genotypes, it is necessary to validate the diagnostic utility of the real-time RPA assay using other ASFV genotypes in further studies. Additionally, this assay was evaluated with the highly virulent ASFV genotype I from Congo and genotype II from Russia, which cause an acute type of infections, but not with the attenuated strain such as the genotype I strain from China. However, the fact that the CP204L(P30) gene, the target of the RPA assay in this study, is highly conserved among ASFV strains indicates that this assay could detect both virulent and attenuated ASFV strains. Further evaluation of the performance of the developed real-time RPA assay over a wide range of clinical specimens including blood, tissue extract, spleen, and bone marrow at various stages of infection will advance the application of this assay to help control ASF through rapid and accurate detection of ASFV.

## 4. Materials and Methods

### 4.1. ASFV DNA Specimens

ASFV DNA specimens were provided by the Federal Research Center for Virology and Microbiology (FRCVM), Volginskii, Russia (Table 4). Because the recent ASFV strains that occurred in Asia were closely related to Georgia/2007 (NC_044959.2) [3], ASFV strains belonging to genotype II and genetically close to Georgia/2007 were selected for use in this study. ASFV DNA specimens were obtained from ASFV field isolates and collected from pigs experimentally challenged with virulent ASFV; the specimens included both ASFV genotypes I and II. Viruses were propagated using primary swine macrophages, which were prepared from pig blood and cultured in RPMI 1640 medium (Gibco, Waltham, MA, USA) supplemented with 30% (*v*/*v*) autologous porcine plasma, 10% (*v*/*v*) fetal bovine serum (Gibco), and antibiotics at 37 °C with 5% CO_2_. The virus titer was determined using tissue culture infectious dose (TCID_50_) assays as previously described [26].

All animal experiments were approved by the Research Ethics Committee of the Federal Research Center of Virology and Microbiology, Russia (No3/2021) and were conducted in accordance with Russian legislation. Large white piglets were obtained from a commercial pig farm in Russia. The pigs were not vaccinated against any infectious diseases and were bred under biosafety level 3AG conditions in the animal facilities at the Federal Research Center of Virology and Microbiology (FRCVM). The piglets were 2–2.5 months in age (weighing 15–18 kg) and were diagnosed as free of specific pathogens (data not shown). Three piglets were injected intramuscularly with 1 × 10^3^ TCID_50_ of ASFV for each genotype for the ASFV challenge. The ASFV Volgograd-v strain was isolated from outbreaks in Russia. The strain belongs to genotype II and is closely related to the reference strain “Georgia/2007 (NC_044959.2)”. The virulent ASFV Congo-v strain (strain K49, MZ202520.1) was isolated from Africa in 1949. It belongs to genotype I and causes an acute form of ASF. Blood samples were collected from the jugular vein of pigs at 3, 4-, 5-, 6-, or 7-days post-infection. DNA was extracted from the infected cell culture and blood samples using a blood and tissue kit (Qiagen, Hilden, Germany) according to the manufacturer’s instructions. The extracted DNA was stored at −80 °C until use.

### 4.2. Synthesis of Standard DNA Samples

High-performance liquid chromatography-pure oligonucleotides corresponding to an amplification target sequence of the *CP204L* gene were custom-synthesized by Integrated DNA Technologies Inc. (IDT Inc., Coral Ville, IA, USA). The synthesized target sequence of *CP204L* was 183 base pairs (bp) in length: (2P30, 5′-TCA TCT TCA AAA CGG ATT TAA GAT CAT CTT CAC AAG TTG TGT TTC ATG CGG GTA GCC TGT ATA ATT GGT TTT CTG TTG AGA TTA TCA ATA GCG GTA GAA TTG TTA CGA CCG CTA TAA AAA CAT TGC TTA GTA CTG TTA AGT ATG ATA TTG TGA AAT CTG CTC GTA TAT ATG CAG GGC AAG GGT-3′). DNA oligonucleotide was dissolved in DNase/RNase-free distilled water to a concentration of 100 pmol/μL, equivalent to ~5 × 10^10^ DNA molecules. The copy number of the synthesized DNA was calculated according to “Calculation for determining the number of copies of a template” (http://cels.uri.edu/gsc/cdna.html, accessed on 5 August 2021) [27]. The synthesized standard DNA was aliquoted and stored at −80 °C until use.

### 4.3. Primers and Exo-Probes

Nucleotide sequences of the ASFV strains from GenBank were aligned to identify the conserved regions of the *C204L* gene. Primers and exo-probes were designed based on the conserved region of the *C204L* gene according to the appendix of the TwistAmp^®^ reaction manual (www.twistdx.co.uk) (accessed on 22 June 2021). Primers and exo-probes with reference sequences of ASFV genotypes were designed, and the probe contained tetrahydrofuran flanked by a dT-fluorophore and dT-quencher. The set of primers and probes that respond to ASFV DNA are shown in Table 1. Primers and exo-probes were synthesized by Macrogen Co. (Seoul, Korea) and IDT, Inc. (Coralville, IA, USA), respectively.

### 4.4. Real-Time RPA Assay

Real-time RPA reactions were performed in a total volume of 25 μL using the TwistAmp™ RT exo kit (TwistDX, Cambridge, UK). The reaction mixture included 0.56 pM of each RPA primer, 0.12 pM exo probe, 19 mM magnesium acetate, and 1 μL of viral or sample DNA template. All reagents, except for the viral template and magnesium acetate, were prepared in a master mix, which was distributed into 0.2 mL tubes. One microliter of viral DNA was then added to each tube. Subsequently, magnesium acetate was pipetted into the tube lids, the lids were carefully closed, and the tubes were centrifuged. After brief vortexing and centrifugation, the tubes were immediately placed in a T-8 device (Axxin, Fairfield, Australia) to start the reaction at 39 °C for 20 min with mixing after 4 min of amplification. The fluorescence signal was measured in real-time and markedly increased upon successful amplification.

### 4.5. qPCR

qPCR was performed using the VDx ASFV qPCR kit (Median Diagnostics, Chuncheon, Korea) on a CFX96 PCR system(BioRad Lab. Ltd., Hercules, CA, USA) according to the manufacturer’s instructions. The reactions were performed as follows: 50 °C for 2 min and 95 °C for 10 min, followed by 40 cycles of 95 °C for 15 s and 58 °C for 60 s.

### 4.6. Performance Assessment of the Real-Time RPA Assay

To analyze the detection limit of the real-time RPA assay, a series of 10-fold serial dilutions of the synthesized standard DNA was prepared to achieve DNA concentrations ranging from 1 × 10^6^ to 1 × 10^0^ copies/reaction. A volume of 1 μL of each DNA dilution was used as the template. The real-time RPA assay was performed in triplicate using standard DNA. Threshold time was plotted against the corresponding amount of DNA detected.

ASFV DNA specimens containing ASFV genotypes I and II from FRCVM, Russia were used to evaluate the sensitivity of the real-time RPA assay (Table 4). In addition, the detection was investigated at timepointspost-infection by using specimens collected from animals challenged with ASFV. To evaluate the specificity, other viruses were used as templates for the assay. A commercially available qPCR kit (VDx ASFV qPCR kit, Median Diagnostics, Chuncheon, Korea) was used to compare the performance of real-time RPA.

### 4.7. Statistical Analysis

Data were analyzed using GraphPad Prism (ver. 6; GraphPad Software, San Diego, CA, USA) program. The sensitivity, specificity, positive predictive value (PPV), and negative predictive value (NPV) of the assays were determined using the number of viruses or specimens obtained from FRCVM as a reference. Differences between experimental groups were analyzed using the Student’s t-test and were considered significant at *p*-values less than 0.05.

## 5. Conclusions

We developed a real-time RPA assay targeting the CP204 gene to rapidly detect ASFV genotypes I and II. This real-time RPA assay could detect ASFV within 15 min, including virus isolates and clinal specimens from pigs challenged with virulent ASFV genotypes I and II strains.

## Figures and Tables

**Figure 1 pathogens-11-00439-f001:**
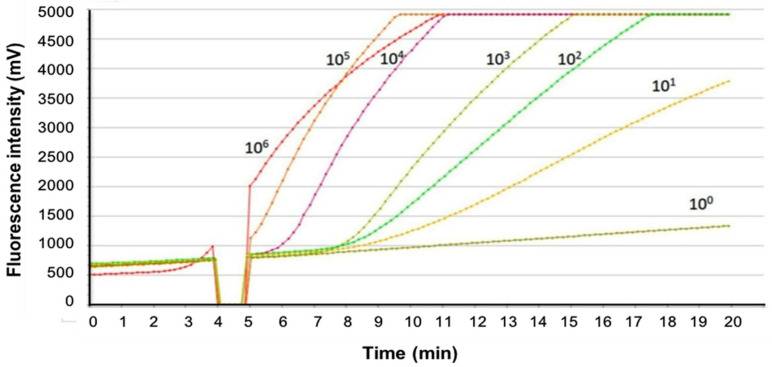
Analytical performance of the real-time RPA assay for ASFV. The synthesized DNA standards in concentrations ranging from 5 × 10^6^ to 5 × 10^0^ copies were amplified as templates for the real-time RPA assay. A representative of three experiments is shown.

**Figure 2 pathogens-11-00439-f002:**
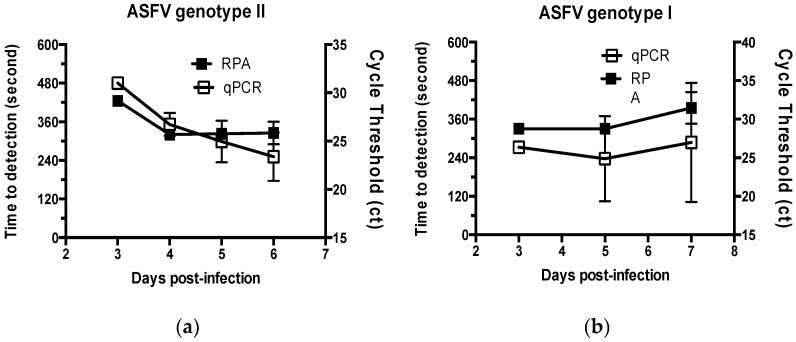
Comparison of the real-time RPA assay and qPCR for ASFV isolates and clinical specimens. The time to detection of the real-time RPA assay and qPCR were measured using11 ASFV isolates and clinical specimens collected from pigs challenged with ASFV (**a**) Volgograd-v (genotype II) or (**b**) Congo-v (genotype I). The time to a positive result for the tested specimens is expressed in seconds.

**Figure 3 pathogens-11-00439-f003:**
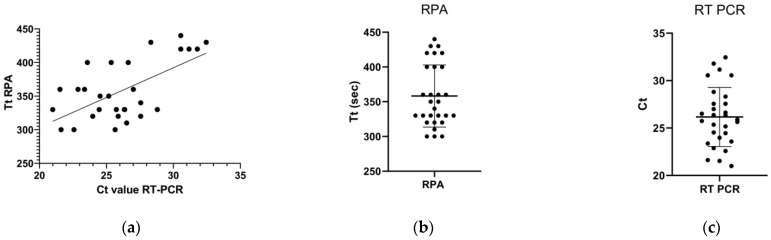
Results of the real-time RPA assay with clinical specimens from pigs challenged with ASFV genotypes I and II. Comparison between the performance of real-time RPA and qPCR on ASFV DNA samples (**a**). Ct values for real-time RPA and qPCR are shown in (**b**,**c**), respectively. Large white piglets were challenged with 1 × 10^3^ TCID_50_ of ASFV Volgograd-v (genotype II) or Congo-v (genotype I), and the blood samples were collected at the designated time point after infection. DNA was extracted from the collected blood samples and used as a template for the real-time RPA assay.

**Table 1 pathogens-11-00439-t001:** Sequences of primers and the probe used in this study.

Name	Nucleotide Sequence (5′-3′)	Amplicon Size (bp)
2RPAP30F	AGATCATCTTCACAAGTTGTGTTTCATGCGGGTAG	141
2RPAP30R	CGAGCAGATTTCACAATATCATACTTAACAGTACT	
2P30Exo v1	TAGCGGTCGTAACAATTCTACCGCTATTGA(Fam dT)A (THF)TC(BHQ1-dT)CAACAGAAAACCAAT-C3spacer	

**Table 2 pathogens-11-00439-t002:** Detection of the ASFV genome in clinical samples using real-time RPA and qPCR assays.

Virus/Specimens	Real-Time RPA		qPCR	
	No. of Positives	No. of Negatives	No. of Positives	No. of Negatives
ASFV GII isolates (*n* = 11)	11	0	11	0
ASFV GII animal specimens (*n* = 14)	13	1	13	1
ASFV GI animal specimens (*n* = 6)	6	0	6	0
Other viruses (*n* = 11)	0	11	0	11
Sensitivity	96.8% (95% CI: 83.3–99.9)		96.8% (95% CI: 83.3–99.9)	
Specificity	100% (95% CI: 71.5–100.0)		100% (95% CI: 71.5–100.0)	
Positive predictive value	100% (95% CI: 88.4–100.0)		100% (95% CI: 88.4–100.0)	
Negative predictive value	91.7% (95% CI: 61.5–99.8)		91.7% (95% CI: 61.5–99.8)	

**Table 3 pathogens-11-00439-t003:** Comparison between the real-time RPA assay and qPCR for the detection of ASFV.

		qPCR		
		No. of Positives	No. of Negatives	Total
Real-time RPA	No. of positive	30	0	30
	No. of negative	0	1	1
	Total	30	1	31

**Table 4 pathogens-11-00439-t004:** List of viruses used in the present study.

Name	Isolates/Origin	Genotype/Serotype	Source
D 1	ASFV Stavropol 01/8	II	FRCVM
D 2	ASFV Irkutsk 2017	II	FRCVM
D 3	ASFV Stavropol	II	FRCVM
D 4	ASFV Omsk #1	II	FRCVM
D 5	ASFV Omsk #2	II	FRCVM
D 6	ASFV Nizhny Novgorod #1	II	FRCVM
D 7	ASFV Nizhny Novgorod #2	II	FRCVM
D 8	ASFV Nizhny Novgorod #3	II	FRCVM
D 9	ASFV Nizhny Novgorod #4	II	FRCVM
D 10	ASFV Krasnodar	II	FRCVM
D 11	ASFV Saratov	II	FRCVM
D 12	ASFV DNA from pig # 1-Volgograd-v (3 dpi)	II	FRCVM
D 13	ASFV DNA from pig # 1-Volgograd-v (5 dpi)	II	FRCVM
D 14	ASFV DNA from pig #2-Volgograd-v (3 dpi)	II	FRCVM
D 15	ASFV DNA from pig #2-Volgograd-v (5 dpi)	II	FRCVM
D 16	ASFV DNA from pig #3-Volgograd-v (3 dpi)	II	FRCVM
D 17	ASFV DNA from pig #3-Volgograd-v (4 dpi)	II	FRCVM
D 18	ASFV DNA from pig #3-Volgograd-v (5 dpi)	II	FRCVM
D 19	ASFV DNA from pig #3-Volgograd-v (6 dpi)	II	FRCVM
D 20	ASFV DNA from pig #4-Volgograd-v (4 dpi)	II	FRCVM
D 21	ASFV DNA from pig #4-Volgograd-v (5 dpi)	II	FRCVM
D 22	ASFV DNA from pig #4-Volgograd-v (6 dpi)	II	FRCVM
D 23	ASFV DNA from pig #5-Volgograd-v (3 dpi)	II	FRCVM
D 24	ASFV DNA from pig #5-Volgograd-v (4 dpi)	II	FRCVM
D 25	ASFV DNA from pig #5-Volgograd-v (5 dpi)	II	FRCVM
D 26	ASFV DNA from pig #1-Congo-v (3 dpi)	I	FRCVM
D 27	ASFV DNA from pig #1-Congo-v (5 dpi)	I	FRCVM
D 28	ASFV DNA from pig #1-Congo-v (7 dpi)	I	FRCVM
D 29	ASFV DNA from pig #2- Congo-v (3 dpi)	I	FRCVM
D 30	ASFV DNA from pig #2-Congo-v (5 dpi)	I	FRCVM
D 31	ASFV DNA from pig #2-Congo-v (7 dpi)	I	FRCVM
V 1	Influenza A virus	H1N1	KBPV
V 2	Influenza A virus	H3N2	KBPV
V 3	Parainfluenza virus 2		KBPV
V 4	Parainfluenza virus 4		KBPV
V 5	Adenovirus	Serotype 1	KBPV
V 6	Adenovirus	Serotype 3	KBPV
V 7	Adenovirus	Serotype 7	KBPV
V 8	Enterovirus A	Enterovirus 71	KBPV
V 9	Rota virus A	Type 1	KBPV
V 10	Rota virus A	Type 3	KBPV
V 11	Cytomegalovirus		KBPV

FRCVM: Federal Research Center for Virology and Microbiology, Volginskii, Russia. KBPV: Korea Bank for Pathogenic Viruses, Korea University, Seoul, Korea. Blood was collected at the designated time points (12–31 days) from pigs challenged with virulent ASFV, either the Volgograd-v or Congo-v strain, and DNA was extracted from the blood.

## Data Availability

The original data used for the analyses can be obtained from the authors after approval by the responsible institutions in Korea and Russia.

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
