# Peer review of "Development of a Real-Time Recombinase Polymerase Amplification Assay for the Rapid Detection of African Swine Fever Virus Genotype I and II"

_pathogens, 2022, doi:10.3390/pathogens11040439_

Round 1
Reviewer 1 Report
The manuscript entitled “ Development of a real-time recombinase polymerase amplification assay for the rapid detection of African swine fever virus genotype I and II” describes a development of a rapid, highly sensitive and specific test for detection of ASFV. Since ASFV poses a serious concern for worldwide pig industry and food safety, the subject is of particular importance. The method is based on a recombinase/polymerase activity, nevertheless in my opinion the study lack novelty since the method was described by several authors previously (Fan et al., 2020; Miao et al., 2019; Zhai et al., 2020; Zhang et al., 2021). Please, compare the sensitivity and specifity of the deveopled method with previous reports in the discussion.
English is sufficient, but the manuscript needs to be carefully checked to improve minor grammatical errors. In the last years numerous similar fast diagnostics methods were developed, nevertheless to my best knowledge, none of it is routinely applied by the veterinarians or farmers to early, on-site detection of ASFV, which must still be confirmed by PCR or qPCR to introduce the appropriate biosecurity measures. Nevertheless, the overall merit is correct, when the authors clearly demonstrate the novelty and practical usage of the developed method. Please relate to the following issues:
Line 38: Yes, but in my opinion the recent jump of the genotype II into Northern Italy would be better worth here
Line 39: Likely is not the best word since wild boar migration is the main reason for disease spreading in Europe, excluding few cases of direct transmission by humans…
Lines 54-59: the general meaning is fine, but please rephrase these sentences since the English here is quite hard to understand.
Line 146-146: This opinion is overemphasized, is 1.5-2 hours of qPCR „a long period of time” to confirm any disease?
Line 163-164 : Indeed, but I would suggest adding this information in the introduction, including the p30 nucleotide similarity between selected genotypes. Please add the information regarding the similarity of this region compared to the other genotypes (if applicable), and based on this express the opinion whether the method could be applied also to them.
Line 184: porcine plasma? Plasma collected from the same animal as macrophages?
Table 4: D1-D12 – if the reference sequence is available at genbank, please add the accession numbers.
Table 4: V1-V11 – why were these viruses selected? It would be worth to apply the pathogens which infect pigs and wild boars.
Line 202, line 204: is the whole genomic sequence available at genbank? If so, please add the accession numbers.
Line 206: Except blood, it is worth testing also the other specimen types, like organs or bone marrow from the dead animals, which is especially important in regard to infected wild boars.
Line 220: DNA molecules in what volume?
Paragraph 4.4: the methodology of the assay is quite complicated, there is a need for laboratory equipment like pipettes, tips, vials, isothermal device etc. And even more important a trained Staff. In my opinion, the application uned the field conditions is quite difficult to perform.
Paragraph 4.7 – insufficient information, what statistical tests were used?
Reviewer 2 Report
In this current state of ASF spread globally, besides the absence of a vaccine, the other challenge has been that of timely diagnosis. The authors have done well to carry out research that can reduce the turnaround time on diagnosis. However, I would like to make some observations concerning the methodology.
How many of the piglets were used in the experiment and how were they grouped? There was no reference to experimental and control groups.
How many were given the genotype I and II respectively and what was the outcome?
For how long was the experiment conducted and what types of samples were collected and analyzed?
Why did the authors bleed from 3 dpi and not 2 dpi? How can we say that there was no viremia by day 2 post-infection? And was the result different when the animals were given high or low viral particles?
Is the CP204L gene 183 sequence the fragment where the primer was designed or was it the primer? If it is the entire gene, then there is no need for it to be there.
Round 2
Reviewer 1 Report
The manuscript has beed improved according to the suggestions. In my opinion, it could be published in present form, since it contains the content which may be considered important for a wide audience.
Author Response
Dear reviewer!
Thank you very much for the productive discussion of our work. Your feedback is very important to us.
Undoubtedly, this helped to improve the quality of our article.
Reviewer 2 Report
The authors have improved the manuscript, however, there remain some minor corrections that had to be done with English for example
Introduction line 34: ...belonging to the family Asfarviridae.
Line 42: "threatening and not threating" etc
The hypothesis and weakness of the study were well spelt out.
Author Response
Dear reviewer!
According to your comments, we were sent the article to professional company for additional english correction. Thank you for your careful study of our work and valuable comments.